# Hydrology across Disciplines: Organization and Application Experiences of a Public Hydrological Service in Italy

Alessandro Allodi, Letizia Angelo, Fabio Bordini, Monica Branchi, Elisa Comune, Mauro Del Longo, Giuseppe Nicolosi, Mauro Noberini, Filippo Pizzera, Alessio Pugliese, Giuseppe Ricciardi *, Fabrizio Tonelli, Franca Tugnoli and Enrica Zenoni

Regional Agency for Prevention Environment and Energy in Emilia-Romagna, 40139 Bologna, Italy; aallodi@arpae.it (A.A.); langelo@arpae.it (L.A.); fbordini@arpae.it (F.B.); mbranchi@arpae.it (M.B.); ecomune@arpae.it (E.C.); mdellongo@arpae.it (M.D.L.); gnicolosi@arpae.it (G.N.); mnoberini@arpae.it (M.N.); fpizzera@arpae.it (F.P.); apugliese@arpae.it (A.P.); ftonelli@arpae.it (F.T.); ftugnoli@arpae.it (F.T.); ezenoni@arpae.it (E.Z.)
* Correspondence: gricciardi@arpae.it

**Abstract:** Water is a fundamental resource for human life and nature; flood management, water supply systems and water protection policies are a few examples of equally important disciplines across the whole hydrological cycle. The present work focuses on the creation and sharing of hydrological knowledge within public activities, with regard to materials and methods adopted for developing and supplying hydrological information, suitable to different stakeholders needs, throughout different disciplines and sectors of environment, economy, society, as well as research and analysis. The aim of this work is to better understand the market in order to increase the value of hydrological data, products and services, and to reduce potential gaps and overlapping areas. The method we developed is based on the example of the Hydrological Service of Emilia-Romagna Region, Italy. Institutional, legal and territorial frameworks as well as agency organization, materials, methods, instruments, activities, products and results are briefly described, focusing on those supporting civil and environmental protection, water management, infrastructure design, climate change adaptation and mitigation measures. We discuss the role of a public Administration in interdisciplinary activities, the links between the general background (e.g., territory, society, rules), organizations, actors, resources, tools, processes and results, by highlighting, where possible, a potential starting point for future research studies. Finally, this paper adopts a novel linguistic style, based on an informal format, in order to explore the set-up and follow-up of the Hydrological Service's initiatives, with the final aim of sparking curiosity and building awareness, from different sectors and disciplines, which, ultimately, may benefit from the presented approaches.

**Keywords:** hydrological services; water cycle modeling; weather related risk management; territorial knowledge; climate change adaptation

## 1. Introduction

The present work has the main objective of giving examples of hydrological products supplied as a public service and how they can effectively be used in water-related applications, and the ultimate goal is highlighting the role of hydrological information within a specific territorial context. The main purpose of this article is to facilitate through these practical examples the connection between research and applications in the field of hydrological observation, modeling, design and support for spatial and sectorial planning and management. Therefore, the novelty of our research relies on the description of many real experiences of cooperation, research and technical applications related to interdisciplinary societal, legal, territorial and economic issues.

The proposed research scheme at the level of the whole of Italy is based on the collection of new information, on bringing new ideas across disciplines, on investigating the

contribution of hydrology in decision-making, society and interdisciplinary collaborations. Finally, we propose a broad spectrum of ideas in the field of hydrological security, which includes all aspects of the water cycle and can not be separated from public awareness, education, economic and environmental sustainability in the territorial framework, according to a glocal approach.

### 1.1. Institutional Framework and Organization in the Environmental Sector

In Italy, the Emilia-Romagna Region, within a comprehensive vision of environmental complexity, underlines the need to support monitoring, vigilance and control through planning and management with the aim of achieving an inclusive, fair and sustainable development, in a prevention perspective (Figure 1).

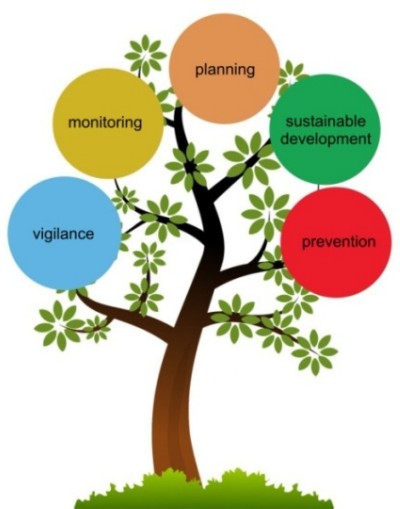

**Figure 1.** Key issues for Arpae, Italy.

Arpae, the Regional Agency for Prevention, Environment and Energy of Emilia-Romagna, supplies information, products, tools and services and supports strategies, plans, programs, projects, training and actions fostering the sustainability of human activities, enhanced by the green economy and the ecological transition, linking private and public initiatives and promoting the territorial capital.

Arpae's work is implemented through the Three Year Programme and Performace Plan, in coherence with the Organization Act and the Services Chart [1].

With its 1.185, among technical and administrative employees and managers, Arpae is part of the National System for Environmental Protection (SNPA), which establishes the essential levels of environmental technical services (in Italian "Livelli Essenziali delle Prestazioni Tecniche Ambientali", LEPTA). The overall effort of Arpae, in terms of full time equivalent per year (FTE/year), for the LEPTA implementation is shown in Figure 2.

The Idro-Meteo-Climate-Structure of Arpae (Arpae-SIMC) carries out operational activities, processes, projects and applied research in meteorology, climatology and hydrology.

The Hydrological Service of Arpae-SIMC (HSA-SIMC) supplies hydrological observations, measurements, hydrological models together with analyses, predictions, and reporting; the Service is composed of 14 experts from different disciplines (IT, geology, engineering, biology, physics), working in two interoperating groups: the monitoring unit and the modeling unit. The groups are coordinated by the service head and supported by the upper level position.

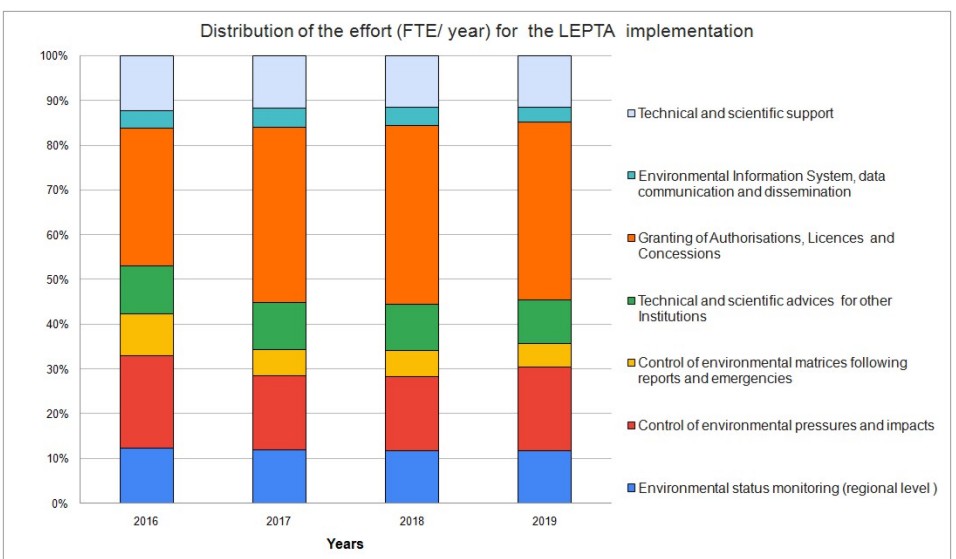

**Figure 2.** Distribution of the effort for LEPTA implementation in Emilia-Romagna.

### 1.2. Legal Framework and Policies of the Water Sector

Water policies and management require distributing specific roles and fostering coherence through cross-sectorial coordination of different levels of governance and stakeholders, and to share policy-relevant water information [2].

Multi-level governance is a guiding and widely recognized principle for the implementation of cohesion policies and, together with the principle of subsidiarity, they represent the basis of the functioning of the European Union [3].

Water policies execution can result in a complex process for either the implementation of measures, which ultimately delivers effects on many other sectors and involves different disciplines, or the different subjects targeted or affected by measures themselves.

Water policies are often based on a multi-actor/multi-layer governance system, where competences and activities are distributed among different territorial and sectorial institutions [4], as well as on the principles that all water is public and that water itself is a good of general interest. Italian water governance includes the EU, national, interregional, regional and local level.

At the EU level, water management and protection are disciplined by Water Framework Directive 2000/60/CE (WFD), WFD Common Implementation Strategy and COM (2012) 673 "A Blueprint to Safeguard Europe's Water Resources". Water quality and quantity are intimately related within the concept of "good status" of water bodies [5]; among all, the hydrological regime is one of the elements used for the classification of the ecological status of water bodies. The River Basin Management Plans, introduced for the first time at EU scale by the WFD, address in a comprehensive manner all the challenges faced by EU waters, pointing out clearly that water management is much more than just water distribution and treatment. Indeed, it involves multi-scale land-use management that affect both water quality and quantity, and it requires coordination throughout all involved authorities in participated spatial planning and definition of funding priorities. Climate change and territorial pressures are global scale phenomena threatening the sustainability of Europe's waters use, and, in particular, its long-term availability, which ultimately represents a real challenge for water managers dealing with quantitative aspects. Pertinent legislation also considers ground water, drinking water, bathing water, sea water, urban wastewater, floods, nitrates, priority substances, REACH (Registration, Evaluation, Authorisation and Restriction of Chemicals), pesticides, biocides, pharmaceuticals and solid waste management.

At the national level, water management and protection are disciplined by the legislative decree 152/2006 (in Italian "decreto legislativo", d.lgs.), which represents the Italian

derivative of the WFD transposition. The decree introduces two main water management tools: (1) the river basin plans, including the water management plans, and (2) the water protection plans, based on hydrological information. In addition, the d.lgs. 30/2009, which implements the EU Directive 2006/118/CE, plays a role in water planning since it deals with the protection of ground water against pollution and deterioration.

At the EU level, flood risk management is disciplined by the EU Directive 2007/60/CE, known as the "flood directive", delivering guidelines on flood risk management plans, based on flood hazard and flood risk mapping.

At the national level, flood management is disciplined by d.lgs. 152/2006, together with d.lgs. 49/2010, which represents the Italian derivative of the flood directive transposition. The Hydrogeological Settlement Plans and the Flood Risk Management Plans are based on the river basin characteristics and on hydrological information, required in all phases referred to prevention, protection, preparedness, flood prediction and early warning. Flood early warning is regulated by two directives of the President of the Council of Ministers promulgated on 27 February 2004 and on 8 February 2013, concerning the institution and operational guidelines for the organization and management of the national distributed early warning system for hydrogeological and hydraulic risk for civil protection. According to d.lgs. 1/2018, civil protection activities related to floods include two main tasks: (a) risk prediction, consisting of the identification and analysis of potential risk scenarios, according to the procedures of either the early warning system or the civil protection planning; (b) risk prevention, aimed at avoiding or reducing the probability of damages occurring as a consequence of disasters, is based on informed knowledge derived from risk prediction analyses and is divided into structural and non-structural activities. Regarding non-structural measures, risk prevention activities are subdivided into several actions: (i) early warning, real time monitoring, surveillance of hydrological events and the evolution of their risk scenarios; (ii) civil protection planning, training, education and updates on new technical rules; (iii) dissemination of risk knowledge, which promotes the awareness and resilience of communities capable of adopting auto-protection measures; (iv) information to a population exposed to disaster risk and (v) civil protection exercises, whereas structural measures in risk prevention mainly include implementing national and regional guidelines, programming and designing flood risk mitigation works, and integrating non-structural and structural actions.

Hydrological activities and procedures carried out by national and regional agencies are addressed in the Decree of the President of the Republic (D.P.R.) 85/1991.

At the regional level, Regional Law (Legge Regionale, LR) 13/2015 of Emilia-Romagna Region [6] updates the territorial government system through the definition of a new role throughout all institutional levels, by identifying new steps of multilevel governance and by reinforcing tools for shared decision-making and acting on political strategies. In the case of Emilia-Romagna, the main actors of interest for water and flood management are: (1) the Emilia-Romagna Region, (2) the Regional Civil Protection, (3) Arpae, (4) the Interregional Agency for the Po River and (5) the Po River District Authority.

*1.3. Territorial Framework of Emilia-Romagna*

The Emilia-Romagna region extends over an area of 22,450 km$^2$; it is located in northern Italy and is part of the Po River District (Figure 3).

Emilia-Romagna can be divided into three main areas: the mountain-hill area (southern and western area); the floodplain area of the Po river (northern and central area); and the coastal area (eastern area).

The main regional rivers can be divided into three main groups, depending on their hydrological and geomorphological characteristics: (1) the Po River tributaries, (2) the Reno river and (3) all the rivers in the Romagna area. Common characteristics among all the three groups are that all of them flow downstream following one principal direction, i.e., from south/southwest towards north/northeast, and all are embanked rivers. Typically, embankments start from the beginning of the floodplain to either the confluence at the

Po River, in case of group (1), or to the river outlet at the Adriatic Sea, in the case of groups (2) and (3). They are all characterized by an intermittent hydrological regime, with strong interannual variability. Higher discharges generally occur from the late autumn to early spring, and they are generated by the most significant meteorological events. Lower discharges occur from late spring to early autumn and frequently they are close to null flowrate, especially downstream alluvial fans. The total length of the primary regional river network is about 600 km, with yearly mean discharge, averaged across all regional rivers, of about 227 $m^3$/s. The total length of the secondary river network is about 1900 km, while the whole hydrographic water network extends for about 13,000 km of natural streams and for 11,500 km of artificial drainage channels. Regional natural lakes are very small and their effect on streamflow regime is negligible, while artificial reservoirs, used for hydropower, agriculture, irrigation and flood control, included in the regional hydrological network, may have an impact on streamflow variability, depending on the season. Total potential reservoir volume in artificial lakes is about 100 million $m^3$, 40 out of 100 belong to the Suviana reservoir, in the Reno River basin. Along the coastline, transitional waters extend over 200 $km^2$. The regional ground water system includes unconfined, semi-confined and confined aquifers.

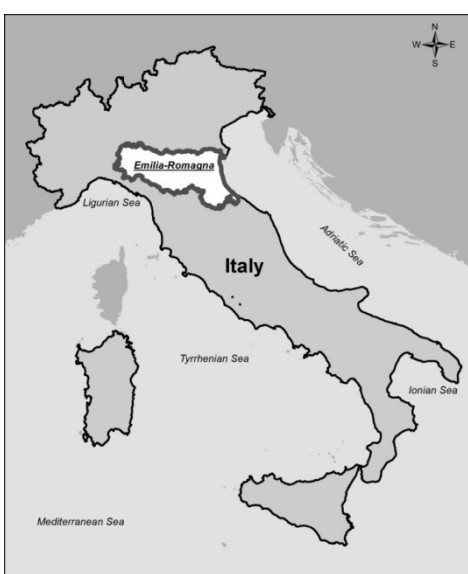

**Figure 3.** The Emilia-Romagna Region, Italy.

Land use areas in Emilia-Romagna Region are distributed as follows: 6200 $km^2$ are covered by forests and 1800 $km^2$ are for agriculture, while urban areas count 328 municipalities, 34 with more than 20,000 inhabitants. Protected areas are about 16% of the total regional territories and include two National Parks, one interregional park, 14 regional parks, 17 national reserves, 15 regional reserves, 158 sites included in "Natura 2000" network (UE Interest Sites—SIC; Special Protection Zones—ZPS). Emilia-Romagna has about 4.5 million inhabitants with a mean population density of 200 inhabitants/$km^2$, with 11.6 million in temporary population for tourism [1].

## 2. Materials and Methods

The HSA-SIMC is in charge of controlling, supporting and delivering technical and scientific responses different actors and stakeholders within the territories of the Emilia-Romagna Region for the management of flood and drought events as well as sectorial and territorial planning for short-, mid-, and long-term time scales. In this section, the main instruments, procedures, methods and activities implemented at the HSA-SIMC are described.

### 2.1. Monitoring

The availability of records of river flow is vital to developing our understanding of the hydrological cycle. Monitoring flowrates at discrete points on a river system allows hydrologists to quantify the integrated output of all hydrological processes acting upon a catchment, and thereby underpins effective water management across areas such as flood risk estimation, water resources management (withdrawal, hydroecological assessment and hydropower generation). Recognizing the fundamental importance of hydrometric monitoring in enabling informed operational and policy decisions across many areas of society led to a growth in river gauging networks throughout the world in the second half of the 20th century [7].

In Emilia-Romagna, a water level gauge exists with one of the longest time series: Pontelagoscuro (FE) on the Po river, where systematic observation started at the beginning of the 19th century [8].

The Emilia-Romagna regional hydrological monitoring network, managed by Arpae SIMC, includes in situ hydrometric observations as well as in situ and remote meteorological observations.

In situ hydrometric observations are mainly obtained from water level gauges coupled with staff gauges (Figure 4); they may sometimes be increased through additional mobile water level sensors.

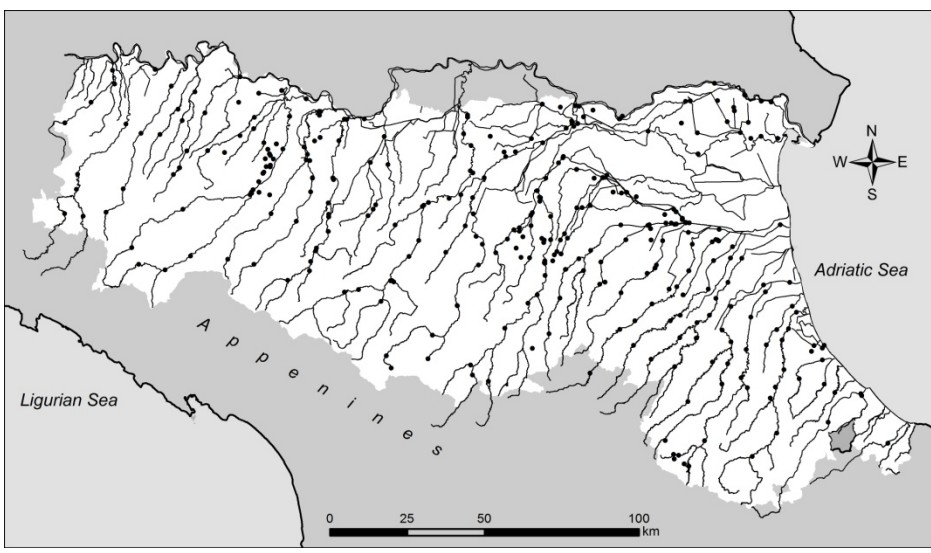

**Figure 4.** Telemetry water level gauges network of Emilia-Romagna.

In situ meteorological observations are mainly obtained from rain gauges, temperature sensors and snow depth sensors; weather stations may also be equipped for the measurement of other variables (for example, solar radiation, humidity, pressure, wind). Arpae-SIMC is also equipped for automatic atmosphere radio sounding.

Almost all the hydrometric and weather stations are included in a telemetry network.

Remote meteorological observations are supplied by weather radars, generating, for example, reflectivity fields from precipitation structures, and satellite products, generating, for example, snow cover estimation and vegetation health status estimation (NDVI).

### 2.2. Field Measurements

Filed measurement mainly concerns liquid discharge measurement and a topographic survey of hydrometric stations.

Liquid discharge measurement, done according to the ISO 9001:2018 Quality Management System, is mainly performed through Acoustic Doppler Current Profilers (ADCP) and Acoustic Doppler Velocimeters (ADV).

Radar surface velocity sensors, salt dilution and other sensors and methodologies are also used for liquid discharge measurement.

Through hydrometric observations and liquid discharge measurements, stage–discharge equations for significant river sections are carefully maintained.

Solid transport measurement, turbidity monitoring (mainly through sample analysis and multi-parameter sondes) and salt intrusion monitoring campaigns in the Po river delta are also included among field measurements.

### *2.3. Modeling*

Hydrological modeling is mainly implemented through the operational Flood Early Warning System (FEWS) and the operational Drought Early Warning System (DEWS, [9]).

FEWS supplies flood predictions through numerical simulation undertaken by hydrological-hydraulic modeling chains fed by meteorological (rainfall, temperature) hourly observations and short- to medium-term forecasts (Figure 5).

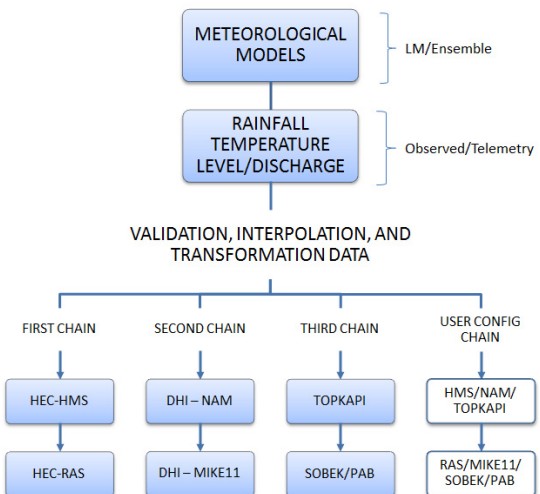

**Figure 5.** Conceptual model of the Flood Early-Warning System (FEWS).

DEWS supplies low flow predictions through numerical simulations undertaken by a hydrological-water balance modeling chain fed by meteorological daily observations and long term forecasts, also considering water withdrawals from the surface water network, water releases from lakes/reservoirs into the surface water network and exchanges between surface and ground water.

Other hydrological modeling tools are available in Arpae SIMC for operational flood prediction and flood management, for soil water balance, agriculture demand prediction, ground water modeling and water quality modeling (including oil spill simulation).

Besides numerical models, statistical models are available for Hydrological Frequency Analysis (HFA) of precipitation, river regimes, flood extremes and low flow extremes. Data driven models are also available and mainly used for flood early warning.

Hydrological observation and modeling are used for populating many indicators and indices, for example the Standard Precipitation Index (SPI) and the Standard Runoff index (SRI).

### *2.4. IT Infrastructure*

A complex IT interlinked infrastructure, including FEWS and DEWS, data bases, GIS application and web services, supports the hydrological activities of the HSA-SIMC. FEWS and DEWS share a significant part of architecture and computational resources. They may work linking data and models for producing real-time forecasts; in this case, operator clients may log the systems to upload updated forecasts. Otherwise, they may run in isolation in the single-user mode on the hydrologist workstation, which is referred to as

stand-alone mode [10]; this solution is suitable for modifying system configuration and for on-demand runs, as, for example, climate-driven hydrological projections, flood scenarios (levee breaks) and low flow scenarios (changing in withdrawals and releases). FEWS and DEWS architectures are based on the optimization of computational resources, through HC parallelization of computationally intensive tasks, and on business continuity.

*2.5. Activities*

Hydrological Service is involved in institutional activities, agreements, working groups and projects, developed to support specific, often complex strongly interconnected processes, in collaboration with Arpae internal actors and external actors, in coherence with the Arpae Three Year Programme and Performace Plan. The main external actors are: the Emilia-Romagna Region, the Regional and National Civil Protection, the National Institute for Environmental Protection and Research, the Interregional Agency for the Po River, the Po River Basin District Authority, Universities and Research Centres and other project partners.

The main institutional activities are related to: hydrological observation, data management and reporting; liquid discharge measurement and stage–discharge equations maintenance; purchase of instruments and equipment; support to modeling implementation and management; support to the Regional Flood Early Warning Center; issuing of technical Advices, for example, the Technical Advice on hydrological analysis of dams and reservoirs, according to the D.P.R. 1363/1959 and to the D.M. 26 June 2014; support, mainly through discharge measurements, to the controls related to the minimum vital flow and to the Water Abstraction Licences; support to the Regional Climate Observatory; support to Strategies, Plans, Programmes, Projects and Actions.

The main agreements are related to: FEWS and DEWS management; FEWS maintenance and support to the Flood Forecast Center for the Po river; hydrological analysis for management of water retention basins during flood events; research, services and products related to operational meteorology, hydrology and sea modeling; targeted hydrological monitoring, discharge measurements, data supply and reporting for the National Water Balance and support to the Observatory on Water Uses in the Po River Basin.

The main working groups are: the National Board for operational hydrological Services; the Regional Group on data sharing; the Regional Group for flood management related to water retention basins; the Regional Group implementing the Strategy for Climate Change Mitigation and Adaptation; the Territorial Laboratory on Smart Cities.

The main (European, national and regional) projects are related to: hydrological statistics for flood management and water resources management, sediments and contaminants transport, integration of surface and ground water modeling; climate change hydrological impacts, disaster risk management, integrated land use and water planning, integration of remote and in situ hydrological monitoring; habitat studies and water quality studies; education to sustainability; application of climate change projections to river discharge analysis [11] and application of the System for Economic and Environmental Accounting for Water (SEEA-Water) [12].

## 3. Results

The proper combination of efforts, resources and synergies invested in institutional, activities, agreements, working groups and projects generates the results described in the following.

*3.1. Hydrological Knowledge*

Building hydrological knowledge is the main result of monitoring, field measurement, modeling and frequency analysis; it is a territorial heritage, a pillar for environmental assessment, river analysis, water and flood risk management (Figure 6), design and management of hydraulic structures.

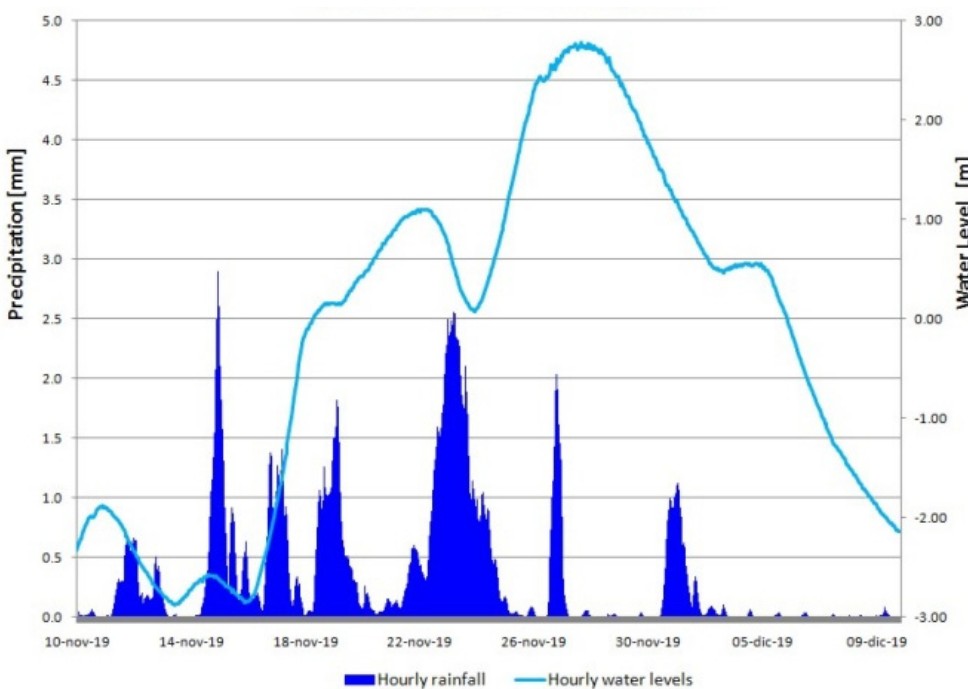

**Figure 6.** Po River at Pontelagoscuro. Observed mean areal precipitation and water level during a flood event in 2019.

Updating of hydrological time series and statistics is requested for the definition of water quantity indicators and indices, water balances, solid transport and pollutant transport. Hydrological analysis of the results given by climate-driven hydrological modeling may assess the influence of climate change and land use on the variability and the statistics of flood and low flow extremes' indicators.

Time and spatial scales as other technical specifications derive from the discussion among hydrologists and the beneficiaries of hydrological knowledge, aimed at balancing costs and benefits in relation to the required applications.

### 3.2. Flood Management

Hydrological information and analysis are supplied by HSA-SIMC within the whole cycle of flood risk management. The main products provided are precipitation and discharge time series, which represent the main outputs from the hydrological modeling chains; these outputs support flood hazard, risk mapping and frequency analyses [13–16]. The results produced also support protection and prevention measures included in the Hydrogeological Settlement Plans, Flood Risk Management Plans, Civil Protection Plans and their implementation programmes, as well as regulations and projects at local, basin and regional scales.

Through the FEWS system, HSA-SIMC provides predictions for flood early warning, surveillance, monitoring and reporting for the Emilia-Romagna Region and the Po river (Figure 7).

FEWS maintenance includes both the management of computational processes and the support to the development and validation of the whole flood modeling cycle, including: hydrological data supply, hydrological analysis, updating of topographic information, hydrologic and hydraulic model calibration and validation, operational implementation and test of hydrological-hydraulic chains; integration of hydrological and meteorological products; assessment and verification of modeling results and of early warning activities and updating of early warning procedures, methods, tools (for example updating of early warning thresholds).

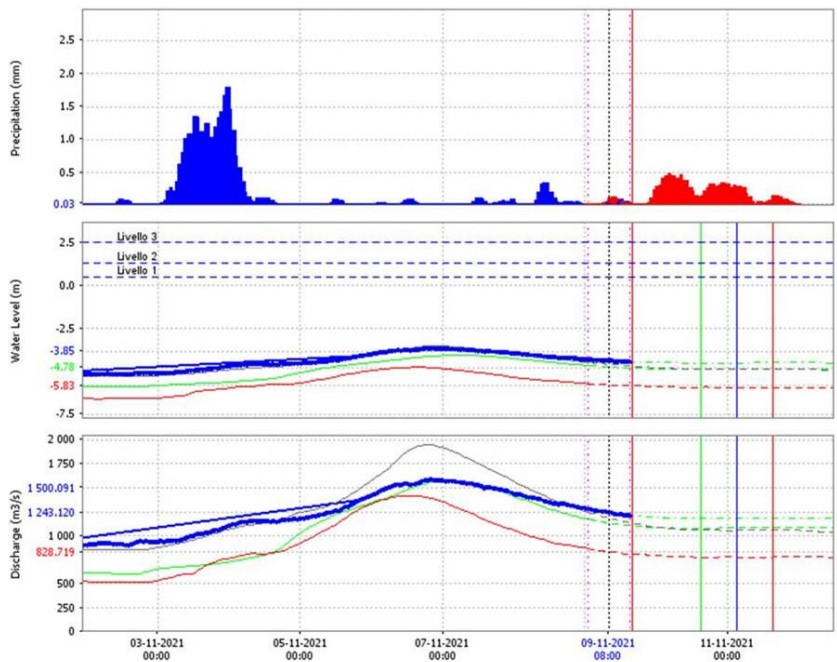

**Figure 7.** Po River at Pontelagoscuro displayed in FEWS System. Observed (blue histogram) and forecasted (red histogram) mean areal precipitation; observed water level and discharge (solid blue line); simulated historical water level and discharge (grey, green and red solid lines); forecasted water level and discharge (grey, green and red dotted lines).

*3.3. Reporting*

Hydrological reporting regarding the most relevant hydrological variables, e.g., snowmelt, floods, low flows, oil spill and others, is related both to periodic publications and to special publications. The list of reports issued in a 5-year period, from 2016 to 2020, is reported in Table 1.

**Table 1.** Report published between 2016 and 2020 including hydrological information.

| Publication | Number | Type | Frequency |
|---|---|---|---|
| Hydrological Yearbooks | 5 | Periodic | Yearly |
| Hydro–Meteo-Climate Report | 4 | Periodic | Yearly (since 2017) |
| Environmental data Web-book | 4 | Periodic | Yearly (since 2017) |
| Climate Observatory Report | 60 | Periodic | Monthly |
| Discharge Bulletin | 1200 | Periodic | Daily (working days) |
| Precipitation/discharge Bulletin | 60 | Periodic | Monthly |
| Minimum Vital Flow Bulletin | 200 | Periodic | 2 days a week Irrigation season (Apr-Sept) |
| Regional flood prediction Bulletin | 1360 | Periodic | Daily (working days + alert days) |
| Po river flood prediction and monitoring Bulletin | 100 | Special | Po river flood events |
| Po river basin water scarcity Bulletin | 50 | Special | Po river low flow periods |
| Flood post-event report | 5 | Special | Flood events |
| Low flow post-event report | 2 | Special | Low flow events |

In addition, HSA-SIMC is involved in the processes of data supply to a variety of remote databases, for European and Italian agencies, as well as river basin and regional authorities. Moreover, HSA-SIMC is in charge for on-demand data supply.

*3.4. Water Management*

Hydrological information and analysis are also supplied for water management including low flow and water scarcity issues.

Precipitation, dry periods and discharge time series, hydrological modeling and frequency analysis support drought hazard assessment, the assessment of the status of surface water bodies, protection and prevention measures and regulations included in Water Protection Plans, Water Management Plans and in their implementation programmes and projects at national, river basin, regional and local scales.

Through the DEWS system, HSA-SIMC supports low flow early warning, prediction, surveillance, monitoring and reporting for the Emilia-Romagna Region, also within river basin-scale activities related to water use management.

DEWS maintenance includes: participation in the management of computational processes; support for the development and validation of hydrological and water balance models and integration of hydrological and meteorological products for low flow and water balance simulations. A specific module is included in DEWS for early warnings related to salt intrusion in the Po river delta.

Moreover, an information and decision support system, based on a GIS platform, has been deployed integrating a telemetry water level network, discharge measurements and mobile water level sensors; this tool gives updated information, available via the web, on the assessment of the presence of the minimum vital discharge in the regional rivers and on the consequent rules for water users, according to the Water Abstraction Licences.

*3.5. Partnership, Dissemination and Interactions with General Public, Stakeholders and Experts*

The Hydrological Service of Arpae SIMC has developed partnerships with other Hydrological Services, Environmental Agencies, Land Reclamation Boards, Provinces, Research Institutes and Universities, Water Companies, NGOs and Engineering Companies.

The dissemination of hydrological activities and the interaction with general public, stakeholders and experts is mainly developed through communities of practices, institutional website, newsletters, press releases, social media, videos, articles, posters, presentations, questionnaires, interviews and meetings.

## 4. Discussion

In the Emilia-Romagna Region, environmental, territorial, agricultural and industrial values, pressures and opportunities are strictly connected, and this sparks different water-related interests and issues, with each other potentially in conflict. The close coexistence of different water needs, i.e., drinking water, agricultural and industries activities, urban areas, protected natural areas and cultural heritage, leads to the opportunity for an integrated approach to flood risk, water and land use management, involving different sectors, actors, processes and tools, with a clear allocation of competences and responsibilities.

In Italy, scientific research in the field of hydrology is mainly carried out by universities, foundations and research centers. The Italian Hydrological Society (IHS) [17] has the main office in Bologna, Emilia-Romagna and promotes the advancement, valorization and dissemination of hydrological sciences with the main objective of bringing together three important realities: the academy (e.g., university and the National Research Council, CNR, CIMA research Foundation), institutional authorities (territorial public bodies, district authorities, civil protection bodies … ) and private operators (Engineering companies, engineers, experts … ). Arpae, in this panorama, behaves as a bridge between public authorities, territorial agencies, and research, providing up-to-date services and data, contributing to solve water-related problems at hand.

Hydrological information is required in the whole legislation and regulation cycle as well as in planning, designing, implementation, monitoring and control processes; in particular, hydrological data and elaboration are fundamental for knowledge-based decision support tools, aimed at effective inter-institutional action, correctly involving and engaging general public, stakeholders and experts.

HSA-SIMC priorities for action are listed as follows:

- the fulfillment of institutional duties and the execution of activities included in Arpae Three Year Programme and Performace Plan, through a proper organization, coherent with governance processes and their consequent declination in project management practices;
- correct understanding of hydrological processes and of their evolution related to hydraulic infrastructures, river morphology, land use, water use and climate change forcing [18];
- better institutional and accountable answers, which trigger the needs from water policies, plans, management and action, especially those related to early-warning and monitoring;
- timely, objective, correct, scientific and authoritative supply source of information and products to fit high standard quality, according to stakeholders needs, and in coherence with established communication and data management plans;
- fair, effective and balanced approach to science, technology and ICT according to the National Digital Agenda;
- creating economies of scale derived from supplying hydrological information suitable for inter disciplinary and cross-sectorial applications.

There are already a few examples of successful partnerships in different projects between HSA-SIMC and other stakeholders where the agency have actively participated in providing technical advice, hydrological data and know-how on hydrological modeling. Hereafter, we report a brief summary of the role of the agency within these projects.

In the FP7 Enhance project, hydrology has supported a multi-risk approach, fostered by Public Private Partnership, integrating economic, environmental and social issues. Two of the project case studies are respectively related to flood, earthquake and climate change risks in densely populated, highly developed land reclamation areas and to flood, drought, water scarcity, erosion and salt intrusion risks in the Po river Delta.

In Interreg Proline-CE Project [19], hydrology has supported the implementation of a Transnational Strategy and of Decision support tools [20] for integrated water and land use management, at different spatial and temporal scales, involving different sectors, aimed at drinking water protection, taking into account flood, drought and other climate driven risks, useful to trigger public participation, in order to correctly allocate knowledge, responsibilities and duties and to best involve actors, stakeholders and experts. The Best Management Practices for the implementation of the strategy include: FEWS, within the Flood Forecast Center for the Po river; DEWS, within the Observatory on Water Uses in the Po Basin; climate change driven flood and drought scenario analysis in the Po river basin. These tools will be useful to select, share and implement optimal actions in order not only to prevent damages but also to prevent and mitigate social conflicts that can potentially arise for water uses between upstream and downstream areas of river basins, but also in case of droughts and flooding events. Best Management practices were designed and tested through workshops, questionnaires, meetings and technical visits.

In the H2020 Clara project [21], hydrological information supported: the co-design and co-development of prototypes and business models for water knowledge and climate services; the demonstration of the added value of climate forecasts versus only observation/climate driven decision support systems and the definition of policy briefs for water and climate change adaptation. Based on the FEWS/DEWS concept, the prototype Service Clara PWA (Parma river basin Water Assessment [22]), based on observations and modeling, was developed for supporting integrated management of water quantity, solid transport, water quality and habitat suitability under Climate Change conditions and different water allocation scenarios. PWA has been designed, implemented and tested according to the needs, potential resources, proposal and requirement coming from market analysis, technical instructions and business plans shared in Multi-User Forums.

In the Interreg BoDEREC-CE Project [23], hydrological information supported knowledge, monitoring, modeling and management of emerging contaminants, especially Phar-

maceutical and Personal Care products (PPCP), in the water environment, delivering decision support tools for water institutions and managers and a comprehensive Transnational Strategy (TRAST-PPCP) carrying legislative, technical, organizational and management recommendations and solutions for future challenges in water security and health protection. The BoDEREC-CE innovative approach was contemporarily adopted in Pilot Areas selected in seven Countries of Central Europe, in order to consider a wide range of different territorial, economic and environmental conditions, delivering a deep knowledge of PPCP occurrence, behavior and fate in ground, karst, surface water and in drinking water treatment plants.

Within the National Mirror Copernicus Programme, hydrological analysis is supporting the implementation of an innovative, open, scalable, interoperable and enabling infrastructure for supplying institutional services based on the Copernicus European Programme for Earth Observation. An interaction methodology [24] has been adopted for the collection and comparative analysis of the needs expressed by the Main Institutional Users (Buyers Groups) and of what can be offered by the research system and the industry; consequently, a set of Accompanying Actions has been developed, aimed at accelerating the development and the competitiveness of, among others, integrated water and climate services, on the basis of an improved and targeted geospatial and in situ observation and models.

Hydrological support to the Emilia-Romagna Territorial laboratory on Smart Cities has focused on the role of water monitoring and modeling in the research field of urban planning, influencing decisions about spaces, geometry, materials and about the role of lakes, channels and rivers within beautiful, climate-proof and resilient Cities [25]. Hydrological scientific and technical support in the design and management of flood retention basins comprise the prediction of flood events at given return periods in different climatic scenarios by means of hydrological models, capable of transforming present and future precipitation forcings, i.e., historical and RCP scenarios, into river streamflows, allowing for an overall evaluation of climate change impacts on flood peak statistics.

## 5. Conclusions

The issues related to the correct implementation of water policies highlight the need for a better interaction between historical, scientific, economic, societal and environmental factors in order to address biodiversity protection and climate change adaptation. According to the European Green Deal, communities, also including small, local ones, will aim for a fair, inclusive and prosperous society, with a modern, resource-efficient and competitive economy; moreover, political and management processes may benefit from specific participatory tools.

In the last years in Emilia-Romagna, due to the economic growth, social progress together with the occurrence of severe weather extreme events, we have been assisting with an increasing awareness, which often times translates into pressure, deriving from public and private uses of water resources, climate change, ecological loss and land use change impact. At the same time, an increasing need for safety from extreme events, including floods and droughts, has been arising, especially within the most exposed and vulnerable communities. Flood risk perception is also related to social, cultural, and psychological factors; low flow events and progressive river degradation may impact local tourism and recreational activities, as construction sites impact territorial accessibility for long periods. Spontaneous grass-root initiatives from large sectors of society are more and more interested in best practices of territorial management.

Hydrological information can contribute to maximizing benefits of water protection, flood risk mitigation and water management and hydrological ecosystem services [26]; coupled with socio-hydrology, it may inform policy by developing a generalizable understanding of phenomena that arise from interactions between water and human systems [27], generating efficiency of stakeholder participation, societal dissemination, transparency,

education to sustainability, experts involvement, public action and synergies with private sectors and NGOs.

The present paper cannot pretend to give a comprehensive representation of the technical and institutional subjects involved, instruments and processes. It is instead intended to be a dissemination work, based on the experience gained in the covered topics, hoping to be used as a potential starting point for further analysis by those who may be interested in the methodological approaches, here represented in a simplified version. Given examples concerning the role of a public Administration and of hydrology information in water management may inspire other similar Agencies to interact with their constituent communities for data generation and interpretation. Potential research on the production, sharing and use of territorial knowledge fostering prevention and sustainability is warmly hoped for.

Further discussion can arise with organizational structures, processes and tools linking scientists, institutions, industry and society in order to cooperate together on legislation, planning, technology, funding and management for sustainability. In addition, cooperation between the public Administration and research is beneficial, especially for climate change mitigation and adaptation measures, flood risk management, ecosystems protection and water management, to cope with uncertain water demand and the transition to more resilient water regimes.

Enforcement of research can also derive from useful exchanges with institutions and society in terms of contribution in co-production of knowledge, policies and implemented measures.

Finally, social, economic and environmental approaches can also be useful to turn research into impactful actions aimed at preventing and mitigating legal and reputational risks for Public Administrations.

Hydrological information is a pillar for science, environment, economy and society. Activities carried on by the HSA-SIMC are based on the awareness of the value of each colleague as a human being, on the sensibility to the environment and to planet life; skills, expertise and attitudes, together with the importance of sharing experiences, which is fundamental for teams. Based on our experience and daily work, we can say that, in the field of water, concerning the proper allocation of resources, the organization and management of weakly-connected processes, the active participation of public institutions, citizens, stakeholders and experts, the decision support tools, the use of living labs and pilot actions and the capitalization of transnational, interdisciplinary experience, a great deal of progress has been made, but there is still some way to go.

**Author Contributions:** Conceptualization, A.A., M.B., E.C., A.P., G.R., F.T. (Fabrizio Tonelli), F.T. (Franca Tugnoli) and E.Z.; formal analysis, L.A., F.B., M.D.L., A.P., F.P., F.T. (Fabrizio Tonelli) and E.Z.; investigation, A.A., L.A., E.C., M.D.L., A.P., F.P., F.T. (Fabrizio Tonelli) and E.Z.; data curation, A.A., M.B., M.D.L., M.N., A.P. and F.T. (Franca Tugnoli); methodology, A.A., M.B., E.C., M.D.L., A.P., F.T. (Fabrizio Tonelli), F.T. (Franca Tugnoli) and E.Z.; resources, A.A., F.B., E.C., M.D.L., G.N. and M.N.; visualization, M.B. and G.R.; software, F.B., G.N. and F.T. (Fabrizio Tonelli); Validation, A.A., M.B. and E.C.; writing—original draft, G.R.; writing—review & editing, A.A., E.C. and A.P.; project administration, A.A. and E.C.; supervision, A.A. and E.C. All authors have read and agreed to the published version of the manuscript.

**Funding:** This research received no external funding.

**Informed Consent Statement:** Informed consent was obtained from all subjects involved in the study.

**Data Availability Statement:** Hydrological data supporting the reported results are available at: https://www.arpae.it/it/temi-ambientali/meteo/report-meteo/annali-idrologici (Emilia-Romagna Hydrological Yearbooks web page, accessed on 2 February 2022).

**Acknowledgments:** The authors thank all technicians and partners involved in the aforementioned institutions, agreements, working groups and projects, contributing with dedication and passion to the maintenance of territorial knowledge and management. The authors thank all colleagues of Arpae and Arpae-SIMC, and all the people who have previously worked at HSA-SIMC.

**Conflicts of Interest:** The authors declare no conflict of interest.

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
