# Peer review of "Hydrology across Disciplines: Organization and Application Experiences of a Public Hydrological Service in Italy"

_climate, doi:10.3390/cli10030032_

Round 1

Reviewer 1 Report

The authors present an overview of the Regional Agency for Prevention, Environment and Energy of Emilia-Romagna, Italy. They have updated the manuscript and addressed the issues raised by the reviewer. There are still a few typographical errors in the manuscript (e.g., last line of p.1-Performance; last line p.11-Po) and a few capitalisation issues (mainly "River" when speaking of a specific basin). The paper seems ready for publication; the strength of the paper is that it presents a model that could be replicated by other governments involved in environmental monitoring and management.

Author Response

Dear Reviewer 1,  thanks for your work on our manuscript and for your feedback.

 We have removed typographical errors including those in the last line of p.1-Performance and in the last line p.11-Po;

We have removed double use of  “river” and  “basin” when speaking of a specific basin.

Best Regards,

Giuseppe Ricciardi

Reviewer 2 Report

Acceptable.

Author Response

Dear Reviewer 2,

thanks for your work on the manuscript and for your comments and suggestions.

Concerning the research design, we agree with the Reviewer so we have added  the following text in the Conclusions:

[Also, cooperation between the public Administration and research is beneficial, especially for climate change mitigation and adaptation measures, flood risk management, ecosystems protection and water management, to cope with uncertain water demand and with the transition to more resilient water regimes].

Concerning the description of methods, Section 2)  can be considered just an essential list of sub sections synthetizing different topics; we think to confirm the text including references for further investigations.

Best Regards,

Giuseppe Ricciardi 

Reviewer 3 Report

1. In the Introduction, there is no information about the purpose and objectives of this study.
2. I did not understand the novelty of your research. Authors need to think it over carefully and convey their main new ideas to the reader.
3. How does your research scheme position itself at the level of the whole of Italy? Is it some unique research scheme? What new things do you bring with your study to Italy's hydrological research and hydrological security? Please give clear answers in the manuscript.
4. In the Discussion section, there is no subsection devoted to the problems facing hydrological research in the indicated region of Italy. Moreover, since this work claims to have some prescriptive role, it is highly desirable to write about possible ways to overcome these problems.
5. In the manuscript (especially in the title and Abstract), there is no connection with the central theme of the journal Climate. Harmonize your exploration with the climate.
6. The English language of the manuscript needs much improvement.

In addition:
(a) The numbering of the figures in the manuscript is confusing. The last figure (Figure 6) is hard to read. Improve its quality.
(b) The numbering of sections of the manuscript is confusing.
(c) Subsection 2.4. IT infrastructure. "... and my work both for seamless prediction ...". Is this a fragment from a personal report of one of the manuscript's authors?

(d) Missing keywords.

Author Response

Dear Reviewer 3,

Thank you for comments and suggestions.

1.In the Introduction, there is no information about the purpose and objectives of this study.

We agree with the Reviewer. In the revised version of the manuscript we added the following sentence "[The present work has the main objective of giving examples of hydrological products supplied as a public service and how they can effectively be used in water-related applications, the ultimate goal is highlighting the role of hydrological information within a specific territorial context]"

2. I did not understand the novelty of your research. Authors need to think it over carefully and convey their main new ideas to the reader.

We are thankful to the reviewer to point this out clearly. The main purpose of this article is to facilitate through these practical examples the connection between research and applications in the field of hydrological observation, modelling, design and support to spatial and sectorial planning and management. Therefore, the novelty of our research relies on the description of many real experiences of cooperation, research and technical applications related to interdisciplinary societal, legal, territorial and economic issues. In the revised version of the manuscript we clarified these aspects both in the Introduction and Discussion sections.         

3. How does your research scheme position itself at the level of the whole of Italy? Is it some unique research scheme? What new things do you bring with your study  to Italy's hydrological research and hydrological security? Please give clear answers in the manuscript.

The proposed research scheme at the level of the whole of Italy is based on the collection of new information, on bringing new ideas across disciplines, on investigating the contribution of hydrology in decision making, society and interdisciplinary collaborations. Finally, we propose a broad spectrum of ideas in the field of hydrological security, which includes all aspects of the water cycle and can’t be separated from public awareness, education, economic and environmental sustainability in the territorial framework, according to a glocal approach. We invite the reviewer to check Introduction in the revised version where we addressed the point he/she arose.  

4. In the Discussion section, there is no subsection devoted to the problems facing hydrological research in the indicated region of Italy. Moreover, since this work claims to have some prescriptive role, it is highly desirable to write about possible ways to overcome these problems.

We thank the reviewer for this advice. In the Discussion section we added the following sentences: "In Italy, scientific research in the field of hydrology is mainly carried out by universities, foundations and research centers. The Italian Hydrological Society (IHS) [17] has the main office in Bologna, Emilia-Romagna and promotes the advancement, valorization and dissemination of hydrological sciences with the main objective of bringing together three important realities: the academy (e.g. university and the National Research Council, CNR), institutional authorities (territorial public bodies, district authorities, civil protection bodies...) and private operators (Engineering companies, engineers, experts ...). Arpae, in this panorama, behaves as a bridge between public authorities, territorial agencies, and research, providing up-to-date services and data, contributing to solve water-related problems at hand."

5. In the manuscript (especially in the title and Abstract), there is no connection with the central theme of the journal Climate. Harmonize your exploration with the climate.

In brief we have extended the abstract including the following text:”

Institutional, legal and territorial frameworks as well as agency organization, materials, methods, instruments, activities, products and results are briefly described, focusing on those supporting civil protection, water management and design, environmental assessment and protection, climate change adaptation and mitigation measures

6. The English language of the manuscript needs much improvement.

We have reviewed  the text according to this suggestion.

In addition:
(a) The numbering of the figures in the manuscript is confusing. The last figure (Figure 6) is hard to read. Improve its quality.

Thanks for the suggestion.  We have reviewed  the text  and Figure 6  and 7 according to this suggestion.

(b) The numbering of sections of the manuscript is confusing.

We have given the correct sequence of sections and subsections

(c) Subsection 2.4. IT infrastructure. "... and my work both for seamless prediction ...". Is this a fragment from a personal report of one of the manuscript's authors?

Sorry, we have substituted the fragment with this new text: “They  may work linking data and models for producing real time forecasts; in this case operator clients may log the systems  to upload updated forecasts. Otherwise  they  may run in isolation in the single-user mode on the hydrologist  workstation, which is referred to as stand- alone mode [10]; this solution is suitable for modifying system configuration and for on-demand runs, as for example climate-driven hydrological projections, flood scenarios (levee breaks) and low flow scenarios (changing in withdrawals and releases)”.

(d) Missing keywords.

Added the following keywords:

  • hydrological services
  • water cycle modelling
  • weather related risk management
  • territorial knowledge
  • climate change adaptation

Best Regards

Giuseppe Ricciardi

Round 2

Reviewer 3 Report

Thank you for the revision of the manuscript.

This manuscript is a resubmission of an earlier submission. The following is a list of the peer review reports and author responses from that submission.

Round 1

Reviewer 1 Report

Interesting article although basic in content. It is a scientific-based technical compilation.

Reviewer 2 Report

The authors present an overview of the Regional Agency for Prevention, Environment and Energy of Emilia-Romagna (Italy), summarising the functions, activities and products produced by this agency. The manuscript has substantial repetition, with the same functions and products being presented as methods, results and discussion. As an overview of the agency, the paper could be significantly shortened; it does not fit the typical methods, results and discussion format. The information presented is interesting and could inspire other similar agencies to interact with their constituent communities both in terms of data generation and interpretation. As such the content is worthy of publication.

The manuscript contains a great many acronyms, many of which are single use, while others are used multiple times. This makes the manuscript difficult to read for a reader that may not be familiar with the organisational structure of the agency. In many cases, it would be helpful to spell out the acronyms.

The manuscript should be reviewed for standard English usage, especially spelling as the manuscript contains many Italian spellings with, for example, an initial "i" when English uses "hy" as in hydrology. There are also several occurrences of the use of "f" where English used "ph" as in morphology. In the results and discussion, the word "as" appears when it seems that "and" would be more appropriate.

The manuscript should be thoroughly reviewed and refined (probably shortened) prior to further consideration for publication.

Reviewer 3 Report

General comments and suggestions:

The paper’s topic is highly interesting and very useful from the aspect of practical water management. However, the topic is too broad, with the paper only generalising the analysed problems and sticking to the general descriptions of the legal and spatial scope of action and presentation of the regular activities implemented as part of the regular programme of action of Arpae Idro-Meteo-Climate-Structure (Arpae SIMC).

Regretfully, in my opinion the paper in the Climate journal is not an appropriate framework for the dissemination of the authors’ experience gained with the analysed topics which they themselves mention as the purpose of the paper in a remark at its end. Consequently, it is proposed that the paper be rejected or thoroughly revised by analysing in more detail some of its segments, respecting in that process the rules about the structure and content of scientific papers usually published in the Climate journal. Although the paper nominally possesses the standard structure of the scientific papers published in the Climate journal (chapters Introduction, Materials and Methods, Results, Discussion, Conclusions), the contents of those sections of the paper do not correspond to their headings and contain mostly descriptions of professional practice which is undoubtedly at a high level, but is nevertheless presented here without enough analytical and critical elements.

I believe that the paper can in the given form be published as a sort of a synthesised conference paper and that it can at a conference encourage a discussion about the achievements of Arpae SIMC and be a good example of the possible improved performance by some similar organisational structures in other areas. According to available information, parts of this paper have in that way been presented at the EGU General Assembly. However, in the given case, with the topic analysed in this way, in my view as the reviewer the paper is not acceptable for publication in the Climate scientific journal and there is no possibility to improve it properly for a more detailed review of its individual sections.

Reviewer 4 Report

It is my pleasure to review the manuscript: climate-1500208, entitled “Hydrology across disciplines: organization and application experiences of a Public Hydrological Service in Italy ”. This paper adopts a novel dialogue-style format to explore the set-up and follow-up of Arpae SIMC- HS initiatives. However, major concerns were address to the significance and results of this study. Hence, it is a pity, but this study could not be recommended for further consideration of the Climate.

  1. The current introdcution is lack of reference, please improve it with more recent published literatures in last five years
  2. The figure quality needs to be enhanced to improve readability
  3. A more detailed description on the modeling based on some theoretical equations is essential
  4. What is IT infrastructure?
  5. Some specific results with numbers, figures, tables are essential
  6. Please add doi or url links for the references.
  7. This study is more like a project resport rather than a scientfic research. What is the significance of this study? What it brings to the research field? Please clarify it in asbtract,introduction, and discussion